# Neuroendocrine Differentiation of Prostate Cancer—An Intriguing Example of Tumor Evolution at Play

**DOI:** 10.3390/cancers11101405

**Published:** 2019-09-20

**Authors:** Girijesh Kumar Patel, Natasha Chugh, Manisha Tripathi

**Affiliations:** Department of Cell Biology and Biochemistry, Texas Tech University Health Sciences Center, Lubbock, TX 79430, USA; Girijesh.Patel@ttuhsc.edu (G.K.P.); natashachugh@yahoo.com (N.C.)

**Keywords:** Androgen deprivation therapy (ADT), cellular plasticity, metastasis, neuroendocrine differentiation (NED), radiation therapy, therapy-induced neuroendocrine prostate cancer (t-NEPC), tumor microenvironment (TME), castration resistant prostate cancer (CRPC)

## Abstract

Our understanding of neuroendocrine prostate cancer (NEPC) has assumed a new perspective in light of the recent advances in research. Although classical NEPC is rarely seen in the clinic, focal neuroendocrine trans-differentiation of prostate adenocarcinoma occurs in about 30% of advanced prostate cancer (PCa) cases, and represents a therapeutic challenge. Even though our knowledge of the mechanisms that mediate neuroendocrine differentiation (NED) is still evolving, the role of androgen deprivation therapy (ADT) as a key driver of this phenomenon is increasingly becoming evident. In this review, we discuss the molecular, cellular, and therapeutic mediators of NED, and emphasize the role of the tumor microenvironment (TME) in orchestrating the phenotype. Understanding the role of the TME in mediating NED could provide us with valuable insights into the plasticity associated with the phenotype, and reveal potential therapeutic targets against this aggressive form of PCa.

## 1. Introduction

Prostate cancer (PCa) is the most diagnosed non-cutaneous cancer and second leading cause of cancer-related deaths in men in the United States [1]. It is estimated that 174,650 new cases will be diagnosed in 2019 and 31,620 will succumb to the disease by the end of this year [1]. Although the majority of PCa develop in the glandular region and are adenocarcinomas, the disease is found to be highly heterogeneous in its molecular profile and clinical behavior [2]. The human prostate consists of epithelial cells (luminal, basal and neuroendocrine cells with varying characteristics), stem cells, and stroma including fibroblasts, smooth muscle cells, myofibroblasts, blood vessels, autonomic nerve fibers, various immune cells, and extracellular matrix components [3]. Akin to normal prostate development, androgen receptor (AR) signaling—mediated by testosterone and 5α-dihydrotestosterone—is key for the development and progression of PCa to the advanced form of the disease [4,5,6,7]. Although all prostate tumors are initially dependent on AR-signaling, the majority of patients administered with ADT develop an androgen-insensitive form of PCa also known as castration resistant prostate cancer (CRPC)—a hallmark of the advanced disease [8]. Seminal advances in recent years have largely shaped our knowledge of advanced PCa, especially in understanding facets of how PCa responds to androgen deprivation therapy (ADT) and other treatment regimen. An intriguing phenomenon that has gained attention in recent years is associated with the clinical observation of how advanced PCa evades androgen-targeted and other therapies. It has been demonstrated that PCa tumors respond to ADT to drive focal neuroendocrine differentiation of CRPC, also termed as therapy-induced neuroendocrine prostate cancer (t-NEPC) [9,10,11,12,13,14]. During the initial diagnosis of PCa, NEPC is rare and ranges from 0.5–2% of total PCa cases. However, t-NEPC has been reported at a much higher percentage [15,16,17] and recent detailed analyses reveal that the incidence of t-NEPC could range from 17–30% [18,19,20]. It is believed that the use of the more potent ADTs is impacting t-NEPC incidence. The origin of NEPC is debatable and still an ongoing topic of research, however, reports suggest that NEPC originates from the trans-differentiation of adenocarcinoma, CRPC cells, or cancer stem cells [19,21,22,23]. A cartoon model, representative of our current understanding of human PCa progression including t-NEPC, is depicted in Figure 1.

In addition, recent progress has led to a widespread appreciation of the reciprocal communications between PCa cells and the cells of the tumor microenvironment (TME), e.g., mast cells, macrophages, bone marrow stromal cells (BMSCs), and cancer associated fibroblasts (CAFs), and how these interactions are critical in shaping PCa progression including NED [24,25,26,27,28,29,30]. The primary factors that make NEPC the most lethal form of PCa is due to the lack of response of the AR to ADT, in part due to weak or absent AR and prostate-specific antigen (PSA); thereby, presenting a therapeutic challenge in the clinic. However, some studies have found the presence of AR and/or AR-signaling in t-NEPC, thereby adding to the complexity in deciphering the phenotype [20]. The limited therapeutic options for the treatment of NEPC include treatments such as cisplatin, or carboplatin with etoposide [31] or docetaxel with a marginal median survival of 7–15 months [32,33,34].

We focus in this review on our current knowledge of the diverse factors that mediate therapeutic resistance in advanced PCa, including how the TME could influence the establishment of t-NEPC. Although continuous advancements have increased our understanding of PCa including CRPC, there is an unmet need to understand the underlying molecular and cellular mechanisms of t-NEPC development in order to identify novel and effective targets for better clinical management of this phenotype.

## 2. Therapy Resistance in Prostate Cancer

Cancer cells adopt a plethora of mechanisms to escape therapeutic regimen including epigenetic and genetic alterations (Table 1) leading to overexpression and amplification of oncogenes, drug expulsion transporters, and drug metabolizing enzymes [35,36,37,38,39,40,41,42,43,44,45,46,47]. In PCa, various modes of drug-resistance have been reported [48,49,50]. The prevalent treatment modalities for PCa include surgical removal of the prostate (radical prostatectomy), or radiation therapy with or without ADT [51]. Although ADT remains a standard of care (SOC) for metastatic prostate cancer, however, after the initial response to ADT, the tumors relapse to the advanced CRPC form of the disease in the majority of patients [52,53]. This has led to the synthesis of the next generation of ADTs such as enzalutamide and abiraterone. Unfortunately, CRPC develops resistance even to the next generation of ADTs and several lines of evidence suggest that long term treatment with ADTs (androgen inhibitor, enzalutamide or androgen synthesis inhibitor, abiraterone) plays a key role in CRPC progression to neuroendocrine differentiation (NED) of CRPC, also termed as CRPC-NE or t-NEPC [10,54,55]. The resistance to ADT results through a variety of mechanisms including constitutive AR-signaling through AR splice variants, ligand-independent activation of AR, amplification, overexpression and/or mutations in the AR gene [47,49,56]. In addition, glucocorticoid receptors (GR) are reported to be overexpressed in the patients treated with anti-androgen therapy (abiraterone or enzalutamide), and GR expression is suggested to bypass AR blockade due to its functional similarity [57,58,59,60]. In addition, reports suggest that cellular plasticity and genetic reprogramming lead to stem-cell-like characteristics promoting the transdifferentiation of neuroendocrine phenotype that is resistant to ADT [57,61,62]. Moreover, radiation therapy also has been reported to induce ADT resistance and NE differentiation [63,64].

## 3. General Characteristics and Molecular Markers of Neuroendocrine Prostate Cancer (NEPC)

NEPC has been shown to progress rapidly, and metastasize primarily to the visceral organs and tissues [61,75]. However, a few reports have also suggested that NEPC metastasizes to the bone [20,76,77,78]. Common features revealed by tumor staining indicate that NEPC is poorly differentiated, displays rosette, swirl or organoid patterns and possesses basophilic appearance, amphiphilic cytoplasm and prominent nucleoli [65,79,80]. According to the World Health Organization classification 2004, large-cell NEPCs show higher mitotic rate and are positive for one or more NEPC markers, while the small-cell NEPCs share features with small-cell lung cancers possessing lower cytoplasmic amount, hyperchromatic nuclei with few nucleoli or no nucleoli [33,80,81]. Subsequent reports—based on the diverse morphologic features—categorized NEPCs into different sub-types: 1. Adenocarcinoma with neuroendocrine (NE) differentiation; 2. Paneth cell NE differentiation; 3. Carcinoid; 4. Small-cell carcinoma; 5. Large-cell NE carcinoma; and 6. Mixed NE carcinoma-acinar adenocarcinoma [82].

NEPC cells commonly show higher expression of genes that are used as markers for NE differentiation, e.g., synaptophysin (SYP), chromogranin (CHGA) and enolase 2 (ENO2). During the last two decades, several additional marker genes have been reported for NEPC including loss of genes: *RB1, TP53*, classical androgen regulated genes and *TMPRSS2-ERG* fusion, and amplification of *MYCN* among others [62,83,84]. NEPC markers proposed by various research groups is listed in Table 2**.**

## 4. Neuroendocrine Trans-Differentiation (NED)

In PCa, neuroendocrine differentiation (NED) is increasingly being seen as an adaptive mechanism that allows PCa cell populations to evade a variety of therapies. Accumulating evidence now suggests that in addition to ADTs [10,11,120] t-NEPC could be induced by radio [12,64] and chemotherapeutic modalities [121]. In mouse and human PCa, NED has been shown to use similar molecular pathways that are found in the endocrine differentiation of the pancreas [95,122]. However, the exact signaling mechanisms by which NE differentiation occurs are largely unknown and remain elusive, thereby making it a challenge to develop therapeutic interventions. The various possible mechanisms of NEPC development that have been proposed by recent reports are summarized in the following subsections.

### 4.1. NED Induced by AR Targeted Therapies

The widespread use of AR pathway inhibitors and the introduction of new and more potent inhibitors to treat CRPC has increased the incidence of t-NEPC [10,18,97,123]. Among the various factors reported to induce NED in PCa include increase in cAMP levels. It is reported that ADT induces the activation of CREB (cAMP response element binding protein) and promotes NED via G protein-coupled receptor kinase 3, GRK3 [124]. cAMP has previously been reported to modulate the cellular morphology, and induce the production of chromogranin (CHGA), synaptophysin (SYP) in LNCaP cells [125]. Along similar lines, Farini et al., showed that the neuropeptide, pituitary adenylate cyclase-activating polypeptide (PACAP) promotes the increased intracellular levels of cAMP, and enhances cell proliferation through the mitogen-activated protein kinase (MAPK) pathway. However, the chronic stimulation of PACAP induced the sustained accumulation of cAMP and activation of CREB, leading to NE differentiation [126]. Furthermore, PAK4 (p21-activated kinase 4) activated by cAMP elevation is reported to enhance the catalytic activity of CREB, and promote hormone- and chemo- resistance and contributes to NE differentiation [127]. Recently, Zhang et al., proposed that ADT induces the upregulation and activation of oncogenic molecule CREB in androgen-dependent (LNCaP and VCaP)- and NEPC cells (NCI-H660 and 144-13). The authors reported enhanced angiogenesis and NE differentiation by CREB through EZH2 (Zeste homologue 2) activity [53]. Furthermore, they showed that EZH2 represses the expression of thrombospondin (TSP1), an inhibitor of angiogenesis, indicating the role of CREB/EZH2 axis in the development of t-NEPC [53]. Therefore, EZH2 may be a promising target to inhibit NE differentiation which may reverse the lineage switch and restore sensitivity to ADT. Further, secretions from NEPC can support LnCaP grafts (human androgen-dependent tumors) to grow in castrated mice. The NE-secreted proteins bombasin and gastrin releasing peptide were identified as activating nuclear factor kappa light chain enhancer of activated B cells (NF-κB) resulting in the expression of androgen-receptor splice variant 7 (AR-V7) in LnCaP cells [47,128]. Activation of NF-κB in the Hi-Myc mouse model also results in CRPC [129]. This provide a mechanism whereby NEPC systemic secretions can contribute to the microenvironment to regulate both NF-κB and AR-V7 in the androgen-dependent adenocarcinoma to induce progression to CRPC.

The Notch signaling pathway is known to be involved in several cellular processes including cell proliferation, differentiation, angiogenesis and apoptosis [130,131]. Deregulation of Notch signaling pathway (e.g., Notch receptors, 1–4; ligands, jagged-1, jagged-2, and δ-like proteins 1, 3, and 4; and Notch target genes, HEY1 and HES1) is observed in several solid malignancies, including PCa [130]. Different Notch pathway-related genes such as delta like canonical Notch ligand 3 (*DLL3*) Hes family BHLH transcription Factor 6 (*HES6*), deltex E3 ubiquitin ligase 1 (*DTX1*), and jagged canonical notch ligand 2 (*JAG2*) are up-regulated, while others such as Notch 2 and 4 receptors are down-regulated in NEPC, indicating the multifaceted role of Notch signaling pathway in PCa. Moreover, Notch acts as an oncogene in the early stages of PCa by promoting metastatic progression while some reports suggest its role as an onco-suppressor in the advanced stages of the disease [114]. Danza et al., showed that low oxygen tension induces NE phenotype by suppressing Notch1 and Notch 2 expression and suppressing the levels of Hes1 and Hey1 in androgen-dependent (LNCaP) and androgen independent (AR-negative PC3 and DU145) cell lines [132]. Reduced levels of Hes1 enhanced the expression of NE markers in LNCaP cells while they remained unchanged in PC-3 and DU145 cells, suggesting the requirement of androgen sensitivity to NE differentiation [132]. Very recently, Notch pathway protein DLL3 (delta-like protein 3) has been reported to be expressed in ~77% of CRPC-NE, while its expression is reported to be 0.52% and 12.5% in benign prostate and CRPC respectively [114]. DLL3 expression is found to correlate with different NE markers e.g., SYP, CHGA, etc. in the ADT treated patients indicating a role of DLL3 in t-NEPC. Although, DLL3 is not reported to drive NED, it however could serve as a potential target for NEPC. DLL3-targeted antibody-drug conjugate SC16LD6.5 (rovalpituzumab-tesirine) that has shown promising effects on AR-negative NEPC (NCT02819999). Another member of the Notch family, ASH1 (human achaete-scute homolog 1) is reported to be highly expressed and positively correlates with NE phenotype in PCa patients who previously received ADT [108]. The co-localization of ASH1 was found in both normal and neoplastic NE cells with CHGA confirmed by IHC staining, indicating the pivotal role of ASH1 in t-NEPC. Furthermore, Rapa et al. reported the overexpression of ASH1 upon androgen deprivation conditions in LNCaP cells and promoted NE phenotype, while the silencing of ASH1 abrogated the NE effect and adversely affected the cell viability [133]. Previously, Hu et al., showed the overexpression of mouse achaete scute homolog-1, mASH1 in a NEPC mouse model and identified mASH1 as a critical factor in NE differentiation in the mouse prostate [134]. They also confirmed ASH1 immunoreactivity within NE foci in tissues of human prostatic adenocarcinoma [134]. The expression of mouse achaete scute homolog-1 was also shown to be induced in neuroendocrine prostate cancer upon the loss of Foxa2 using transgenic adenocarcinoma of themouse prostate (TRAMP) mouse model [135]. Another Notch family protein, DLK1 (Delta-like non-canonical Notch ligand 1) has been reported to be expressed in the stem-and neuroendocrine cells [136], while DLK1 is suppressed by Notch signaling in the intermediate differentiated prostatic cells. However, Notch-1 was found to be overexpressed in differentiated epithelial cells, indicating hierarchical relationship in stemness [136]. These reports indicate the role of Notch signaling in stemness [137], a common feature in NEPC.

Hypoxia and hypoxia-inducible factors (HIF) are shown to play important roles in several malignancies. Recently, Guo et al., reported the relation between hypoxia and one cut homeobox 2, (ONECUT2), a neuroendocrine driver. ONECUT2 inhibits androgen signaling and promotes NE differentiation in PCa [138]. ONECUT2 expression positively correlates with angiogenesis and hypoxia related genes. Transcriptome analysis of ONECUT2-modulated cells under normoxic and hypoxic conditions revealed 120 genes up-regulated by ONECUT2 and 25 of them are related to hypoxia pathway, indicating that ONECUT2 regulates hypoxia-induced gene expression in NEPC cells by regulating HIF1α binding to the chromatin without affecting HIF-1α protein level. In addition, ONECUT2 was found to activate SMAD family member 3 (SMAD3), a co-factor of HIF-1α thereby synergizing hypoxia signaling and drive NEPC development [138]. In another similar study, it was reported that ONECUT2 suppressed the translational activity of AR and repressed the expression of Forkhead Box A1 (FOXA1) [139]. The authors identified that ONECUT2 also regulates paternally expressed 10 (*PEG10*) by directly binding to the PEG10 promoter. Previously, FOXA1 was shown to be an important factor involved in positioning of AR on chromatin and its transcriptional reprogramming [140]. Another report showed that, PEG10 is a master driver of NEPC [111] and promotes invasion and proliferation of PCa cells with NEPC phenotype. Overall these findings indicate the important roles of ONECUT2 and PEG10 in CRPC progression and NEPC development.

The neuronal transcriptional regulator SRRM4 (Serine/Arginine Repetitive Matrix 4) is highly expressed in patients that exhibit the NEPC phenotype and significantly correlates with NE markers (CHGA and SYP and CD56) expression [45,46]. Zhang et al., reported that the expression of SRRM4 correlated with loss of RE1-silencing transcription factor (REST) or lack of repressor activity in REST’s splice variants due to the absence of transcriptional repressor domain [45]. Recently, Li et al., reported that the alternative RNA splicing of REST is regulated by SRRM4 using RNA-sequencing data [46]. Moreover, they reported that enzalutamide treatment enhanced the expression of SRRM4 in LNCaP and VCaP cells, and induced the NE phenotype. In addition, shRNA-mediated loss of RB1 or TP53 in LNCaP cells further potentiated SRMM4-mediated expression of NE markers, CHGB, ENO2, SYP and SCG3, indicating that SRRM4 as a potent driver of NE differentiation [105]. Recently, Li et al., further demonstrated that NE tumors that trans-differentiated from CRPC showed enhanced expression of SRRM4 and negatively correlated with patient outcome [46].

Recently, Reina-Campos et al., reported reduced expression of PKCλ/ι (protein kinase C λ/ι), encoded by *PRKCI*, in both de novo- and t-NEPC [141]. They demonstrated that the loss of PKCλ/ι enhanced mTORC1/ATF4 axis signaling by phosphorylation of - late endosomal/lysosomal adaptor, MAPK and mammalian target of rapamycin (MTOR) activator 2 (LAMTOR2) leading to increased S-adenosyl methionine synthesis. The increased levels of intracellular S-adenosyl methionine drives the epigenetic changes and promotes DNA hypermethylation to favor NE differentiation [141]. Furthermore, treatment with DNA methyltransferase (DNMT) inhibitor (decitabine) significantly reduced the NEPC phenotype indicating that metabolic reprogramming and DNA methylation could be targeted to inhibit NE differentiation in PCa [141].

The T-type calcium channels (TTCCs) are the low voltage-activated calcium channels. TTCCs have been shown to be involved in embryonic development and progression of several malignancies [142]. Recently, Silvestri et al., reported the overexpression of TTCCs isoform calcium voltage-gated channel subunit alpha1 G (CACNA1G) in advanced PCa that correlates with ADT resistance and poor prognosis while TTCC inhibition resulted in reduced PC3 cell proliferation and survival [143]. Previously, Gackiere and colleagues showed that another TTCC, CaV3.2 was highly expressed in NE cells derived from LNCaP cells. Moreover, the increased activity of CaV3.2 channel is proposed to enhance the secretion of oncogenic factors with a role in disease progression and androgen resistance [144]. Earlier, Vanoverberghe et al., reported that NE cells showed impaired Ca^2+^ homeostasis due to the suppressed expression of SERCA 2b Ca^2+^- ATPase and luminal Ca^2+^ binding/storage chaperone, calreticulin leading to low level of endoplasmic reticulum Ca^2+^ ions [145]. These findings underscore the role of Ca^2+^ levels and calcium channels in castrate resistance and NEPC development.

### 4.2. Other Therapies That Mediate PCa NED

Cyclooxygenase-2 (COX-2), an inducible pro-inflammatory enzyme, known for the conversion of arachadonic acid to prostaglandins and other eicosanoids, has been shown to be overexpressed in several malignancies including PCa [146]. COX2 inhibitors are commonly prescribed drugs for the control of inflammation, and have been demonstrated to possess anticancer activity and reduced risk of cancers [147]. The effects of COX2 inhibitor (NS-398) on LNCaP and C4-2B cells were evaluated and reported to induce apoptosis in LNCaP cells but not in its aggressive and androgen unresponsive subline, C4-2B. Interestingly, instead, due to NS-398 treatment, C4-2B cells underwent an unusual neuroendocrine-like differentiation which was mediated by NF-kB induced expression of macrophage migration inhibitory factor (MIF), a proinflamatory cytokine [146].

In another study, phytoestrogen (genistein)—an isoflavone from soyabean—has been shown to induce NED of PCa cells [148]. Pinski et al., demonstrated that LNCaP cells treated with genistein showed trans-differentiation with NEPC phenotype in the surviving cells with enhanced expression of NE markers such as CHGA, SYP, serotonin, and beta-III tubulin [148]. PCa cells have been shown to be responsive to adrenergic and thyroid hormones [149]. This study demonstrated that beta-adrenergic stimulation (isoproterenol) reduced PCa cell growth and tumor burden in mice but induced CREB signaling and the expression of NE-related proteins including CHGA, ENO2, SYP and pCREB, etc. [149], indicating a role in NE differentiation.

### 4.3. Ionizing Radiation (IR) Induced PCa NED

Ionizing radiation (IR) is the first line of therapy for the localized PCa. Most patients respond well but 10% of low risk and 60% of high-risk patients show cancer relapse within five years of IR therapy [150,151]. IR therapy is documented to induce NE differentiation in PCa [12,64] by increasing the nuclear content of phosphorylated CREB and cytoplasmic sequestering of ATF2. Furthermore, stable expression of non-phosphorylated CREB or constitutive nuclear localization of ATF2 was found to inhibit IR-induced NE-like differentiation. Moreover, the authors also observed that ionizing radiation-resistant de-differentiated cells were resistant to docetaxel and ADT [12,63]. Similarly, Suarez et al. reported increased activation of CREB after exposure to IR and induction of NED, while the knockdown of CREB sensitized PCa cells to IR and inhibited NE differentiation [152].

### 4.4. The Role of TME and Epigenetic Reprogramming in PCa NED

It is now widely appreciated that the various types of cells in the TME exert stimulating effects on PCa cells and vice versa. The cross-talk between the PCa cells and the cells present in the TME have been shown to drive genetic and epigenetic alterations in both the cellular compartments leading to the malignant phenotype, epithelial-mesenchymal transition [153], altered drug efficacy [25,154,155,156], and NE differentiation [30]. The different TME cells includes nerve cells, fibroblasts, lymphocytes, macrophages, endothelial cells, and smooth muscle cells, and are known to exert either pro- or -anti-tumorigenic activities [157]. Some of the TME cells are reported to play roles in NE differentiation of PCa.

Mast cell are being increasingly recognized as critical components in the TME that support PCa growth, angiogenesis, therapy resistance, immune suppression and NE differentiation [26,158,159,160,161]. Recently, Xu et al., showed that protein kinase D (PKD) secreted from PCa cells help in mast cell recruitment in the TME and tumor angiogenesis. Increased activation of PKD was associated with enhanced tube formation by HUVEC cells and mast cell infiltration, while silencing of PKD abrogated the effect significantly [158]. Previously, Xie et al., showed that co-culture of C4-2 and mast cells induce docetaxel and radiotherapy resistance via activating p38/p53/p21signaling pathway [161]. In relation to the role of mast cells in NED, Ou et al., reported that co-culture of PCa cells (LNCaP and C4-2) with mast cells (HMC-1) resulted NE phenotypes in PCa cells by activation of p21 [159]. Furthermore, there is now evidence corroborating the role of mast cells in treatment-induced NED. Dang et al., showed that treatment with enzalutamide enhanced mast cell infiltration in TME via modulating AR/IL-8 signaling and enhanced secretion of chemo-attractants such as IL-8, adrenomedulin and CCL8 [26]. The recruited mast cells suppressed AR expression and self-feedback to recruit more mast cells. This AR suppression enhanced the expression of NE markers (CHGA, SYP and ENO2) via miR-32, indicating the role of mast cells in AR suppression and NE differentiation [26]. It is reported that c-Kit, a tyrosine kinase receptor and its ligand, stem-cell factor (SCF) play an essential role in the development and homeostasis in different tissues [162]. c-Kit/SCF signaling has also reported in the development of several malignancies and in maintaining stemness [163]. Mast cells are shown to express c-Kit and are strongly dependent on SCF ligand [164] for their development and survival [165]. In contrast to other reports, Pittoni et al., showed inhibitory effects of mast cells on NE differentiation [166]. The authors reported that PCa cells secrete SCF that recruits mast cells in TME. SCF is sequestered in the TME by binding to c-Kit receptors on mast cells. The mast cells in the TME produce MMP-9 that enhanced the outgrowth of well-differentiated adenocarcinoma cells by promoting angiogenesis, EMT and invasion. Further, tumor progression leads to origin of foci of poorly differentiated adenocarcinoma that produces its own MMP-9, making the tumor independent of mast cells. Targeting of mast cells releases the sequestered SCF, making it available for binding to c-Kit receptor of prostate stem cells. The enhanced c-Kit/SCF signaling leads to prostate stem-cell proliferation and progression to NE differentiation [166,167]. Similarly, Jachetti et al., demonstrated that targeting of mast cells using imatinib, a c-Kit receptor inhibitor, resulted in partial benefits against prostate adenocarcinoma by inhibiting the supportive mast cells but promoted NE differentiation due to defective signaling downstream of the c-kit receptor [168]. Taken together, the role of mast cell in NE differentiation is controversial and needs further investigation.

Another cell-type present in the TME are macrophages, a kind of white blood cells that engulf and digest microbes, cancer cells, and foreign materials that do not belong in healthy body. They are involved in both innate and adaptive immunity. In the TME, macrophages are modulated by cancer cells to support tumor growth and progression, and are called tumor-associated macrophages (TAMs). Lee at al., have shown that PCa cells secrete BMP-6 (bone morphogenetic protein-6) which induces TAM-derived IL-6 secretion [27]. The secreted IL-6 from the macrophages promotes NE differentiation in both, human PCa cells and TRAMP-C2 cells. Neutralizing the IL-6 or removal of macrophages abrogated the NE phenotype, indicating the role of macrophages in NEPC [27]. IL-6 is well known to regulate tumor growth in several malignancies through the IL-6 receptor (IL-6R) and glycoprotein 130 (gp130/CD130) and activates diverse signaling pathways including MAP Kinase, JAK/STAT, PI3K/AKT, etc. [169,170,171,172]. More recently, Wang et al., reported that treatment with enzalutamide induces HMGB1 (High mobility group protein B1) expression and facilitated TAM recruitment and polarization, and promotes NE differentiation via stabilization of β-catenin [173]. Yu et al., showed that activation of Wnt/β-catenin pathway in a mouse model of PCa promoted NED of the tumor [174]. Activated TAMs secrete IL-6 to enhance enzalutamide-induced NE differentiation by promoting HMGB1 expression through STAT3.

Cancer-associate fibroblasts (CAFs) play multifaceted roles in carcinogenesis by regulating tumor cell proliferation and invasion, via affecting growth factors, cytokines, extracellular matrix (ECM) stiffness, and immune evasion [175]. Recently, Mishra et al., reported that epigenetic reprogramming in CAFs leads to neuroendocrine differentiation promoting interactions within the TME [30]. By whole-genome methylation analysis of prostate fibroblasts, the authors identified epigenetic silencing of a Ras inhibitor (RASAL3) in CAFs cells. The authors found that ADT further enhanced RASAL3 epigenetic silencing and resulted in Ras driven glutamine secretion by CAFs. The secreted glutamine gets taken up by the epithelia and leads to mTOR activation and mediates NE differentiation indicating the Ras mediated metabolic reprogramming in CAF affects NE differentiation [30]. In another study, Kato et al., reported that ADT treatment increased the CD105^+^ fibroblastic population in PCa patient-derived xenograft tissues. CD105 signaling contributes to SFRP1 (secreted frizzled-related protein 1) upregulation in CAFs, which induced NED in the epithelia [155]. However, the stromal induced NED was suppressed when ADT was given in combination with CD105 inhibitor. Another study by Rochette et al., demonstrated an increased expression of asporin (an androgen responsive gene) in the CAFs in response to the surrounding cancer cells. The enhanced expression of asporin in CAFs is shown to be induced by p53 inhibition in mouse models and correlates with NE marker expression in tumor tissues [176], indicating a cross-talk between CAF and epithelial cells in NEPC.

Ammirante et al. showed that ADT induces the expression of CXCL13 in myofibroblast cells that enhanced the recruitment of B lymphocyte in the TME [177]. In response to the hypoxia induced by the ADT, there is an activation of HIF-1α, which induced TGF-β expression. In response to the presence of TGF-β, fibroblasts are converted to CXCL13 expressing myofibroblasts. CXCL13 expression leads to recruitment of B cells. Moreover, blockade of TGF-β signaling or immune-depletion of myofibroblasts, suppress the B-cell recruitment in tumors and prevented the emergence of a more aggressive type of cancer, NEPC [177].

In PCa, bone is one of the preferred sites for metastasis for adenocarcinoma. During bone metastasis, the cross-talk between PCa cells and BMSCs plays a crucial role in cell survival and apoptosis [29,178]. It has been reported that TGF-β secreted from BMSCs induces pronounced apoptosis in the androgen-dependent cells (LNCaP) compared to androgen independent (C4-2) cells. TGF-β signaling in PCa cells leads to NE differentiation [29]. Although primary NEPC rarely metastasize to the bone, in t-NEPC up to 15% of the bone metastasis are NEPC [20]. However, the appearance of t-NEPC within the bone maybe due to a trans-differentiation of existing bone adenocarcinoma rather than new metastasis by NEPC tumors [179].

### 4.5. Exosome-Mediated PCa NED

Exosomes are nanovesicles (30–150 nm) and are known to be involved in intercellular communications either by inducing signals directly via their surface molecules or by the transfer of vital molecules including nucleic acids, proteins and other small molecules, e.g., nucleotides, sugars, etc. to the recipient cells. Encapsulated with different molecules, exosomes are known to be involved in cancer progression, pre-metastatic niche formation [180], organ specific metastasis [181], immunosuppression, and drug resistance [41,182]. They also contain cell-signature molecules that can serve as a potential diagnostic and prognostic markers [183]. Therefore, based on the diverse functions of exosomes it is imperative to explore their role in NE differentiation. Recently, Lin et al., reported the role of exosome in the NE differentiation by transferring the adipocyte differentiation-related protein (ADRP). The expression and packaging of ADRP was induced with the treatment of either IL-6 or ADT to PCa cells. The ADRP was shown to play a key role in PPARγ-induced adiposome accumulation and NE differentiation of CRPC and is associated with increased expression of CHGA and ENO2 [184].

### 4.6. Cellular Plasticity/Phenotype Alteration in PCa NED

ADT resistance and relapse is very common in advanced PCa that relates to the extensive intra-tumoral genetic and phenotypic heterogeneity. Mounting lines of evidence suggest that epigenetic reprogramming leads to PCa progression, drug resistance and NE differentiation [57,74,96,185].

Ku et al., reported that in a mouse model of prostate adenocarcinoma with phosphatase and tensin homolog (PTEN) mutation, RB1 loss facilitates lineage plasticity. Additionally, loss of TP53 in this mouse model led to ADT resistance [74]. Further, the authors performed gene expression analysis of mouse and human tumor tissues and found overexpression of epigenetic reprogramming factors EZH2 and SOX2 and identified their involvement in NE differentiation. The inhibition of EZH2 and SOX2 have been shown to enhance AR expression and sensitivity to ADT [57,74] indicating their role in ADT resistance and NED. RB1 is known to regulate cell cycle by suppressing the transcription factor E2F1. McNair et al., reported that direct RB loss is found to be associated with lethal CRPC development and led to E2F1 cistrome expansion and showed different binding specificity as compared to the one observed due to RB inactivation [186]. Mu et al., reported that lineage plasticity promoted by SOX2 in TP53/RB1 negative PCa caused NED rather than a trans-differentiation of the tumor [57].

Mounting lines of evidence point to the important role of N-MYC in drug resistance and NE differentiation through lineage plasticity [65,70,85,96,187,188]. Lee et al., reported that N-MYC and AKT1 play important roles in driving adenocarcinoma to NED [70]. Very recently, Berger et al., identified that aberrant expression of N-MYC is related to epigenetic reprogramming in PCa and NED [96]. Previously, N-MYC has been shown to induce transcriptional reprogramming via EZH2 and promoted NED [85] indicating an important role of EZH2 in PCa pathogenesis and NE differentiation. Aurora kinase A (AURKA) is overexpressed in NEPC, it stabilizes N-MYC and promotes cell cycle progression and proliferation. The AURKA inhibitor, Alisertib disrupts the interaction between N-MYC and AURKA and suppresses tumor growth significantly [189]. A phase II clinical trial of AURKA inhibitor in patients with de novo or t-NEPC concluded that a subset of patients with molecular features related to AURKA and N-MYC showed significant clinical benefit [79]. Besides N-MYC, the overexpression of c-MYC is commonly reported in PCa. In association with AR, Barefield et al., have reported that c-MYC overexpression partially reprogrammed AR chromatin occupancy and altered its transcriptional program. c-MYC overexpression was also found to be associated with altered histone marks distribution, e.g., H3K4me1 and H3K27me3. Further, c-MYC overexpression suppressed AR target genes, PSA and GNMT (Glycine N-Methyltransferase), indicating the transcriptional reprogramming of AR whereby c-MYC serves as a driving force of aggressive PCa [190]. In another study, integrative analyses of epigenetic and transcriptional landscapes by Park et al., revealed a common set of genetic and epigenetic processes between small-cell prostate cancer (SCPC) and small-cell lung cancer (SCLC). These authors report a set of common oncogenic drivers between SCPC and SCLS that can induce a neuroendocrine cancer in spite of being developed from distinct organs. [122].

FOXA1, a well-studied transcription factor plays an essential role in the development and differentiation of the epithelial cells and its knockdown in mouse models is embryonic lethal with impaired prostatic development [191]. Moreover, the same group showed that expression of FOXA1 is essential for the maintenance of cellular differentiation in the prostate [192]. Recently, Kim et al., found that FOXA1 negatively regulates IL-8 expression by binding to the IL-8 promoter. Loss of FOXA1 leads to the activation of mitogen-activated protein kinases, originally called ERK, extracellular signal-regulated kinases (MAPK/ERK) pathway by enhancing ERK phosphorylation and JAK/STAT3 pathway-related gene expression [87]. Upregulation of IL-8 was shown to induce the expression of NE marker, ENO2 which is negatively correlated with FOXA1 expression [87]. Rotinen et al., demonstrated that ONEOUT2 is responsible for metastatic CRPC survival by suppressing AR transcriptional activity. ONECUT2 negatively regulates the expression of FOXA1, and enhances the expression of genes associated with NE differentiation [139]. Moreover, ONECUT2 is also shown to induce NE plasticity by suppressing AR activity, synergized by hypoxia signaling [138]. Very recently, Adams et al., showed that mutation in FOXA1 (R219) led to the non-canonical motif (GTAAAG/A) enrichment in metastatic tumor with NE phenotype. Moreover, R219 mutation was shown to block luminal differentiation but activated the mesenchymal and neuroendocrine transcriptional program in mouse prostate organoid while wild-type FOXA1 promoted the luminal differentiation. This mutation was reported to correlate with open chromatin at thousands of genomic loci which provide exposed sites of FOXA1 binding and enhance gene expression [193]. Altogether, these findings underscore the role of epigenetic reprogramming, lineage plasticity and NE differentiation in PCa.

Taken together, mounting evidence suggests that NE differentiation of PCa is a complex process that is mediated by a plethora of factors. The different kinds of cells in the TME, and other factors that play vital roles in the induction of NED are depicted in Figure 2.

## 5. Potential Therapeutic Strategies for NEPC

The prolonged use of ADTs has been recognized to induce NED of prostate cancer which currently lacks effective therapeutic modalities. Based on existing studies, platinum-based therapy in combination with docetaxel is the common treatment option for the NEPC [34]. Several therapeutic agents that are currently under clinical trial are listed in Table 3. The clinical management of PCa is recently reviewed and updated in recent publications [194,195,196,197]. Several NEPC molecules that are thought to drive NEPC progression have been reported to date, and are being used in clinical trials for the better management of advanced PCa. In a phase II clinical trial (NCT01799278), alisertib, a drug that disrupts the interaction between N-MYC and AURKA showed exceptional response in a subset of NEPC patients [79]. Another, clinical trial (NCT02963051) in metastatic castration resistant prostate cancer (mCRPC) and NEPC patients is in phase I stage which involves administration of intravenous copper chloride in combination with oral disulfiram (Anatabuse). This therapeutic combination acts by increasing reactive oxygen species (ROS) generation and inhibits DNA methyltransferase activity and ubiquitin-proteasomal pathways [198,199,200]. This combinatorial strategy may be an effective therapeutic regimen in future. A recently completed phase I and II trial (NCT02819999) with DLL3-targeted antibody-drug conjugate SC16LD6.5 (rovalpituzumab-tesirine) resulted in promising effects on AR-negative NEPC. Although immune checkpoint inhibitors have not been successful for the management of PCa, a phase II trial (NCT03179410) at Duke University, with the inhibitor of PD-L1, avelumab is being tested for t-NEPC. Currently in phase III clinical trial is ^177^Lu-PSMA-617- PSMA targeting ligand with chemically attached radioactive atom Lutetium-177. This drug binds to PSMA-positive PCa cells present at multiple sites in the body and releases beta particle radiation to kill cancer cells. Targeting PSMA may result in a better outcome in advanced mCRPC patients including metastasis to the bone.

PCa is a highly heterogeneous disease and a single therapeutic agent is usually not sufficiently effective. Therefore, the clinical management of advanced PCa is being tried in combinatorial therapeutic approaches, among them several are under clinical trials such as cabazitaxel + carboplatin + prednisone + olaparib (NCT03263650), and Radium-223 + dexamethasone (NCT03432949). Metformin has been reported to be cytotoxic to cancer stem cells in several malignancies, including breast, colorectal, pancreatic, and glioblastoma [212]. Furthermore, it has been shown to enhance the efficacy of paclitaxel, carboplatin, 5-fluorouracil, and oxaliplatin, and reduces the expression of cancer stem-cell markers [212], indicating that metformin is a potential drug to target cancer stem cells. Recently, metformin has been shown to inhibit PCa initiation and progression by inhibiting TAMs and is proposed to be a promising regimen when administered in combination with ADT [213].

Besides the ongoing clinical trials, some of the identified molecules which have been implicated in the progression to NEPC can be potentially targeted. Among them, EZH2 is one of important targets and its inhibition in preclinical models has showed promising effects [85]. Another promising target is ONECUT2 which is reported to mediate the progression to NEPC. The treatment with a small molecule inhibitor, CSRM617 against ONECUT2 showed promising effects in PCa xenografts. CSRM617 could serve as an important therapeutic agent for the mCRPC and AR-positive NEPC [139]. Recently, Lee et al., has identified carcino-embryonic antigen-related cell adhesion molecule 5 (CEACAM5) using cell surfaceome analysis as a novel target for the immunotherapy for NEPC [107]. Another molecule, PEG10, highly expressed in NEPC patients, could be a promising therapeutic target against NEPC [110]. Similarly, SRRM4 is highly expressed in NEPC and is associated with loss of REST and serves as a potent driver of NED. SRRM4 specific inhibitor may be a potential therapeutic option to target NED, but the lack of the crystal structure of SRRM4 is a major obstacle. Other molecules associated with NED like SOX2 and BRN2 could be the potential targets for the clinical management of NEPC. Tumor-associated macrophages (TAMs) shown to promotes NEPC via IL-6 and STAT3. Combined inhibition of the IL-6/STAT3 pathway (by Tocilizumab) and HMGB1 (by knockdown) resulted in suppression of enzalutamide-induced NE differentiation in PCa, indicating that dual targeting of IL-6 and HMGB1 may serve as a promising treatment option for enzalutamide-resistant PCa [173].

## 6. Conclusions and Future Perspectives

Despite recent therapeutic advancements in treating advanced PCa, NEPC management is a clinical challenge. The common use of newer, more potent ADTs for AR inhibition (enzalutamide) and AR synthesis inhibitor (abiraterone) has increased the progression free survival to some extent, but increasing evidence now suggests that a large proportion of heavily treated CRPC patients display many features of NED. Although several reports in the recent years have suggested that t-NEPC is about 25–30% of tumors in castration resistant prostate cancer (CRPC) patients, the numbers might be even higher as advanced PCa patients are often not biopsied. It is therefore imperative to identify novel molecular and cellular targets for better treatment and prognosis of t-NEPC in the clinics. Progression to t-NEPC may be an evolutionary mechanism devised by advanced PCa cells to evade therapy. In addition, the roles played by the cells of the TME in this induced plasticity are often ignored, and need to be studied more carefully. Studies that delineate the effects of anti-cancer drugs including ADTs on the cells of TME and their paracrine effects on NED will provide crucial missing pieces of the puzzle. The previously generated genetically engineered mouse (GEM) models of PCa show that loss of p53/Rb in the mouse are important steps for the development of spontaneous NE phenotype, and this molecular mechanism mimics human PCa progression [24,174,214,215,216,217,218,219]. These mouse models can be ideal in unraveling the mechanisms of trans-differentiation of adenocarcinoma to NEPC, and in testing new therapeutic modalities to block the progression and/or in the treatment of NEPC [174,217,218,219]. In summary, along with studying the molecular mechanisms that mediate NED, defining the cellular contributions as well as deciphering the cross-talk between TME and PCa will be cardinal in tackling the NED phenotype.

## Figures and Tables

**Figure 1 cancers-11-01405-f001:**
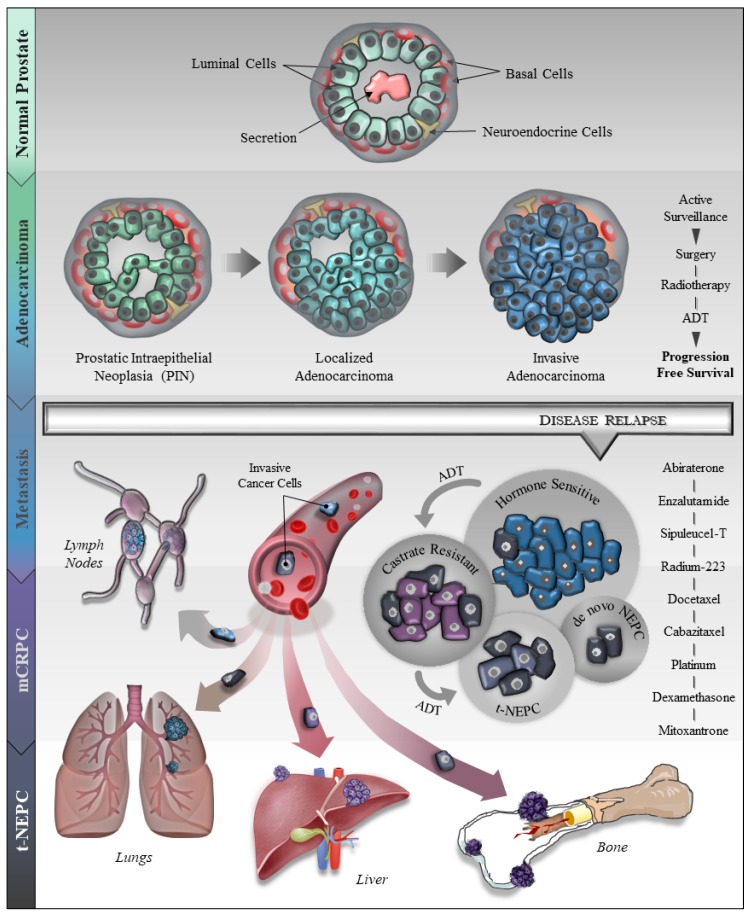
A generalized overview of prostate cancer (PCa) progression, metastasis, drug resistance and neuroendocrine differentiation (NED). The illustration describes PCa development from normal epithelial cells (Basal, Luminal and NE cells) to prostatic intraepithelial neoplasia (PIN) to localized- and invasive adenocarcinoma. The cartoon depicts several therapeutic regimens used for the treatment of PCa including surgical resection, radiotherapy and androgen deprivation therapy (ADT). After the initial response to ADT, the majority of the patients relapse with resistance to ADT leading to castration resistant prostate cancer (CRPC) with or without metastasis. These patients are further treated with the next-generation ADTs, enzalutamide or abiraterone. During the course of CRPC treatment, about 30% of PCa patients develop a more aggressive and fatal form of the disease called t-NEPC that has very limited therapeutic responses.

**Figure 2 cancers-11-01405-f002:**
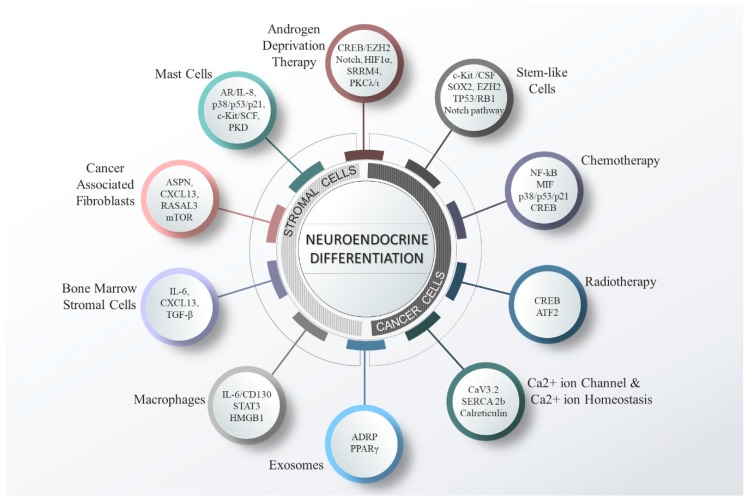
The diverse factors (cellular, molecular and therapeutic) involved in mediating neuroendocrine differentiation (NED) in PCa. Various factors that affect PCa cells include androgen deprivation-, radio- and chemo-therapy. In addition, the cells of tumor microenvironment (TME) including mast cells, cancer associated fibroblasts (CAFs), macrophages and bone marrow stromal cells (BMSCs) have been shown to promote the NED. Furthermore, calcium ion channels and alteration in calcium ion homeostasis play crucial roles in drug resistance and NED. In addition, Exosomes secreted from PCa cells have also been associated with NED. A few examples of molecular pathways are represented.

**Table 1 cancers-11-01405-t001:** Alteration in gene expression (approximate %) in therapy-induced neuroendocrine prostate cancer (t-NEPC).

Gene	Case %	Status	Reference Number
v-myc avian myelocytomatosis viral related oncogene, Neuroblastoma-derived (*MYCN)*	40%	Up-regulated	[10,65]
Aurora Kinase A (AURKA)	40%	Up-regulated	[10,65]
Cluster of differentiation 46 (CD46)	46%	Up-regulated	[66]
Serine/arginine repetitive matrix 4 (SRRM4)	~50%	Up-regulated	[45]
Transmembrane protease, serine 2 *-ETS-related gene (TMPRSS2-ERG)*	~50%	Rearrangement	[67,68,69]
AKT Serine/Threonine Kinase 1 (AKT1)	28%	Up-regulated	[70]
CYCLIN D1	88%	Loss	[71]
Retinoblastoma Protein 1(RB1)	70–90%	Loss	[48,72,73]
Phosphatase and tensin homolog (PTEN)	90%	Loss	[74]
Tumor supressor protein p53 (TP53)	56–67%	Loss or Mutation	[48,72]

**Table 2 cancers-11-01405-t002:** List of proposed markers associated with neuroendocrine prostate cancer (NEPC).

S. No.	Gene Name (Symbol)	Reference Number
**Up-regulated genes**		
1.	Synaptophysin/major synaptic vesicle protein p38 (*SYP*)	[10,81]
2.	Chromogranin A and B (*CHGA/CHGB*)	[10,81]
3.	Aurora kinase A (*AURKA*)	[10,65]
4.	Neuroblastoma-derived v-myc avian myelocytomatosis viral related oncogene (*N-MYC*)	[10,65,85]
5.	Enhancer Of Zeste 2 Polycomb Repressive Complex 2 Subunit (*EZH2*)	[53,74,85,86]
6.	Neuron-specific enolase (*NSE/ENO2*)	[10,81,87]
7.	Calcitonin (*CALC1*)	[88,89]
8.	Secretogranin II (SCG2) and III (SCG3)	[45,90,91]
9.	Vasoactive Intestinal Peptide (*VIP*)	[92]
10.	Gastrin Releasing Peptide (*GRP*)	[93]
11.	NK2 homeobox 1 (*NKX2.1*)/Thyroid transcription factor 1 (*TTF-1*) and NKX2.2	[94,95,96]
12.	Neural cell adhesion molecule (*NCAM1/**CD56*)	[48,97]
13.	Forkhead Box A2 (*FOXA2*)	[98,99,100,101]
14.	*WNT11*	[102]
15.	POU Class 3 Homeobox 2 (*POU3F2*/BRN2)	[103,104]
16.	Serine/Arginine Repetitive Matrix 4 (*SRRM4*) (RNA splicing factor)	[45,46,105]
17.	Sex Determining Region Y (SRY)-Box 2 (*SOX2*) and *SOX11*	[23,57,96,103,106]
18.	Carcinoembryonic antigen-related cell adhesion molecule 5 (CEACAM5) or CD63E	[107]
19.	human achaete-scute homolog 1 (*ASH1/ASCL1*)	[108,109]
20.	Paternally expressed10 (PEG10)	[86,110,111]
21.	TMPRSS2-ERG gene rearrangement	[67,112,113]
22.	P16 or cyclin-dependent kinase inhibitor 2A	[71]
23.	Delta-like protein 3 (DLL3)	[114]
**Gene loss/Down-regulated genes**		
1.	Androgen receptor (*AR*)	[10]
2.	Prostate-specific antigen/ kallikrein-3 (*PSA/KLK3*)	[10]
3.	Retinoblastoma tumor-suppressor gene (*RB1*) and TP53	[48,57,72,74]
4.	Forkhead Box A1(*FOXA1*)	[87]
5.	*PTEN/AKT1*	[72,85,115]
6.	RE1 Silencing Transcription Factor (REST)	[45,116,117]
7.	Tumor suppressor *CYLD*	[48]
8.	SAM pointed domain-containing ETS transcription factor (*SPDEF*)	[48,118,119]
9.	*Cyclin D1*	[71]

**Table 3 cancers-11-01405-t003:** Summary of some ongoing clinical trials in NEPC.

S. No.	Name of the Drug	Target	Trial No.	References
1.	Alisertib (MLN8237)	AURKA	Phase II completed (NCT01799278)	[79]
2.	Intravenous copper and oral disulfiram	Nuclear Protein Localization Protein 4 (NPL4)	Phase Ib (NCT02963051)	[198,200]
3.	Rovalpituzumab-tesirine (SC16LD6.5)	DLL3	Phase I & II completed (NCT02709889)	[114]
4.	Avelumab	PD-L1	Phase II (NCT03179410)	[201]
5.	GSK126, GSK343, fGSK503	EZH2	Preclinical	[85]
6.	^177^Lu-PSMA-617	PSMA	Phase III (NCT03511664)	[202,203,204]
7.	Next-generation AR/pathway inhibitors (Orteronel + Prednisone)	CYP17 lyase	Phase II Completed (NCT01549951)	[205,206]
8.	Seviteronel	CYP17 lyase and AR inhibitor	Phase II (NCT02445976)	[207]
9.	Panobinostat and Bicalutamide	histone deacetylase inhibitor (epigetic pathways)	Phase I & II (NCT00878436)	[208,209]
10.	Cabazitaxel + Carboplatin + Prednisone + Olaparib	DNA recombination	Phase II (NCT03263650)	[210]
11.	Radium-223 + Dexamethasone	mCRPC with bone metastasis	Phase IV (NCT03432949)	[211]

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
