# Peer review of "Neuroendocrine Differentiation of Prostate Cancer—An Intriguing Example of Tumor Evolution at Play"

_cancers, 2019, doi:10.3390/cancers11101405_

Round 1

Reviewer 1 Report

Overall the review is a compilation of many studies linking to various molecular markers with NEPC tumor type. The authors discussed many different mechanisms of neuroendocrine differentiation including the tumor microenvironment. Altogether, the manuscript is worth publishing for researcher in the area of advanced prostate cancer after the authors address the comments below.

Specific comments:

The authors should include a paragraph describing possible clinical utility of the diverse factors (Fig 2) that can have application for disease management. Please discuss the factors, which can be better associated with the disease phenotype.

Please provide a table for various published clinical therapeutic studies (Platinum, aurora kinase inhibitor, and others) in NEPC patients, and include original reference for each clinical study.

Reviewer 2 Report

This is a comprehensive and interesting review and highly relevant for tackling advanced patients with NEPC as well as research strategies to identify molecular and cellular targets for treatment of NEPC. It covers multiple proposed mechanisms to enhance our understanding of NEPC and the figures in particular are an excellent summary. Priorities for research are also addressed and I think this review is worthy of publication. 

Very minor moments:

Is there a better way of describing the table 2 instead of “by research groups”. I’m not entirely sure what research groups means here as they seem to be divided into unregulated and down regulated genes.

Just double check grammatical things throughout manuscript - nothing some copyediting can’t fix. i.e. line 42 page 1 - respond (without the s)

Round 2

Reviewer 1 Report

The authors addressed all the comments in the revised manuscript.